# Is vaginal delivery of a fetus in breech presentation at an extremely preterm gestational age associated with an increased risk of neonatal death? A comparative study

Clémentine Pierre[1,2]*, Audrey Leroy[1], Adeline Pierache[3], Laurent Storme[1,2], Véronique Debarge[1,2], Sandrine Depret[1], Thameur Rakza[1,2], Charles Garabedian[1,2], Damien Subtil[1,2]

1 Univ. Lille, CHU Lille, Hôpital Jeanne de Flandre, Pôle Femme Mère Nouveau-né, Lille, France, 2 Univ. Lille, EA 2694, METRICS, Evaluation des Technologies de Santé et des Pratiques Médicales, Lille, France, 3 Univ. Lille, CHU Lille, Département de Biostatistiques, Lille, France

* clementinepierre@hotmail.com

**Data Availability Statement:** All relevant data are within the paper.

## Abstract

### Background

The effect on neonatal mortality of mode of delivery of a fetus in breech presentation at an extremely preterm gestational age remains controversial.

### Objective

To compare mortality associated with planned vaginal delivery (PVD) of fetuses in breech presentation with that of fetuses in breech presentation with a planned cesarean delivery (PCD).

### Material and methods

Retrospective study reviewing records over a 19-year period in a level 3 university referral center of singleton infants born between $25^{+0}$ and $27^{+6}$ weeks of gestation, alive on arrival in the delivery room, and weighing at least 500 grams at birth. Infants in the first group were in breech presentation with PVD and the second in breech presentation with PCD. The principal endpoint was neonatal death.

### Results

During the study period, we observed 113 breech presentations with PVD, and 80 breech presentations with PCD. Although not significant after adjustment, neonatal mortality in the breech PVD group was more than twice that of the breech PCD group (19.5 vs 7.8%, $P = 0.031$, ORa = 2.6, 95% CI 0.8–9.3, NNT = 8). This higher neonatal mortality in the breech PVD group was exclusively associated with a higher risk of death in the delivery room (12.4 vs 0.0% $P = 0.001$, OR not calculable, NNT = 8). In these extremely preterm breech

**Funding:** The authors received no specific funding for this work.

**Competing interests:** The authors have declared that no competing interests exist.

presentations with PVD, neonatal mortality in the delivery room was associated with entrapment of the aftercoming head, cord prolapse, and a short duration of labor.

## Conclusion

For deliveries between $25^{+0}$ and $27^{+6}$ weeks' gestation, vaginal delivery in breech presentation is associated with a higher risk of death in the delivery room.

## Introduction

Preterm birth is the leading cause of perinatal mortality and morbidity [1, 2].

The conditions of birth of preterm infants can play a determinant role in their immediate and long-term prognosis. This prognosis is best if the birth is preceded by antenatal corticosteroid therapy and takes place in a center appropriate to the extent of prematurity (moderately preterm at 32–36 weeks of gestation (weeks), very at 28–32 weeks, and extremely before 28 weeks) [3–5]. Although mode of delivery does not necessarily appear to be a prognostic factor in preterm deliveries [6]. the situation when the fetus is simultaneously in breech presentation and very preterm is somewhat uncertain [7].

Before 32 weeks of gestation, vaginal delivery may involve risks [8], although they are not unanimously reported [9]; they appear most frequent for extremely preterm delivery [7, 8]. Most authors report that a trial of vaginal delivery for fetuses in breech presentation before 28 weeks is accompanied by an increase in neonatal mortality [7, 10–15], although a few have not reported it in extremely preterm infants [16, 17]. The cause of this possible excess of neonatal mortality is mostly not specified. Few countries recommend a routine cesarean delivery in this situation [18]; others make no recommendations at all about mode of delivery [19, 20].

In view of these gaps and uncertainty in the literature, we conducted a study to measure the potential neonatal excess risk of mortality associated with vaginal delivery of fetuses in breech presentation born extremely preterm. In order to better understand the possible excess mortality in case of trial of labor, total neonatal mortality was separated as i) death during delivery plus ii) death in the neonatal intensive care unit (NICU).

## Material and methods

This retrospective study examined files of patients in our university hospital level 3 maternity ward from the start of 1997 to the end of 2015. Women were eligible for one of these study groups if they had a singleton pregnancy in breech or cephalic presentation and gave birth between $25^{+0}$ and $27^{+6}$ weeks of gestation during the study period by vaginal or cesarean delivery to a liveborn infant with a birth weight of at least 500 g. Other presentations, medically indicated terminations of pregnancy, fetal deaths, fetuses with congenital malformation, and pregnancies involving imminent threats to the mother or fetus justifying an emergency cesarean (e.g., eclampsia or abruptio placentae) were not eligible.

To measure the potential excess neonatal mortality for fetuses in breech presentation born extremely preterm after planned vaginal delivery (breech PVD), we compared them with fetuses in breech presentation born after planned cesarean delivery (breech PCD).

The women's characteristics were collected from their obstetric files: maternal age, smoking, parity, previous cesarean delivery, spontaneous or induced preterm delivery, premature rupture of the membranes (yes/no), corticosteroid therapy (yes/no), duration of labor (from arrival in delivery room to delivery), gestational age at delivery. Mode of delivery was defined

by what had been planned before or at the very start of labor, regardless of outcome. Accordingly, a woman arriving in labor for whom vaginal delivery was judged possible was included in the PVD group, even if a cesarean delivery was finally performed during labor. Data collected about delivery included cord prolapse during labor, entrapment of the aftercoming head/difficulties during delivery and use of forceps. We also extracted data about the neonatal outcome: birth weight, 5-minute Apgar score, both the umbilical artery pH, base deficit and lactates at birth. We also noted the presence (yes/no) in the newborn of intraventricular hemorrhage of grade 3 or 4 on transfontanellar ultrasound, or grade 3 or 4 leukomalacia, whether or not associated with cerebral hyperechogenicity for more than 10 days, and the onset of intracranial hemorrhages. The principal endpoint was total neonatal mortality defined as i) death during delivery plus ii) death in the neonatal intensive care unit (NICU). The use of retrospective data for this study was reported to the French Data Protection Authority (2020–460). Under French regulations, this study is exempt from IRB review because it is an observational study using anonymised data from medical records.

Women are informed that their records can be used for the evaluation of medical practices and are provided the option to opt out of these studies. They have signed an non-objection form for the use of data from their medical file.

Categorical variables are expressed as numbers (percentage). Quantitative variables are expressed as means (± standard deviation, SD) in the case of normal distribution or medians [interquartile range, IQR] otherwise. Normality of distributions was assessed using histograms and the Shapiro-Wilk test. Comparisons of mothers' characteristics according to planned mode of delivery (PVD vs PCV) were performed with Chi-square tests for categorical variables (or Fisher' exact tests when expected cell frequency was <5), Student t or Mann-Whitney U tests (regarding the normality of distributions) for quantitative variables and Cochran Armitage trend tests for ordinal variables. Comparisons in outcomes (newborn conditions) between the two planned mode of delivery (PVD vs PCD) were done using logistic regression models for binary outcomes and linear regression model for umbilical artery pH. Comparisons in neonatal mortality outcome were further adjusted for pre-specified confounding factors (weight, gestational age and gender at delivery). Odds ratios (ORs) for binary outcomes and difference in means for umbilical artery pH were derived from regression models as effect sizes with theirs 95% confidence intervals (CIs).

In infants born in breech planned vaginal delivery (PVD), factors associated with death in the delivery room were assessed using univariable logistic regression models; no multivariable regression model was done regarding the small number of events. Odds ratios (ORs) and theirs 95% confidence intervals (CIs) were calculated as effect sizes. Finally, comparisons in the risk of head entrapment and neonatal deaths in delivery room for breech PVD according to gestational age (25 vs 26 vs 27 SA) were done using Cochran Armitage trend tests. Statistical testing was conducted at the two-tailed α-level of 0.05. Data were analyzed using the SAS software version 9.4 (SAS Institute, Cary, NC).

## Results

During the 19-year study period, our hospital had 90,320 deliveries, including 1,144 (1.3%) singleton deliveries between $25^{+0}$ and $27^{+6}$ weeks of gestation (Fig 1). After the exclusion of 72 women whose fetuses weighed less than 500 grams at birth, 42 presentations that were neither cephalic nor breech, 273 medically indicated terminations of pregnancy, 100 fetal deaths, 18 congenital malformations, 14 emergency cesareans due to imminent threats to the mother or fetus, and 6 missing files, there remained 619 deliveries of live singletons between $25^{+0}$ and $27^{+6}$ weeks of gestation with a birth weight of at least 500 grams eligible for our study. Among

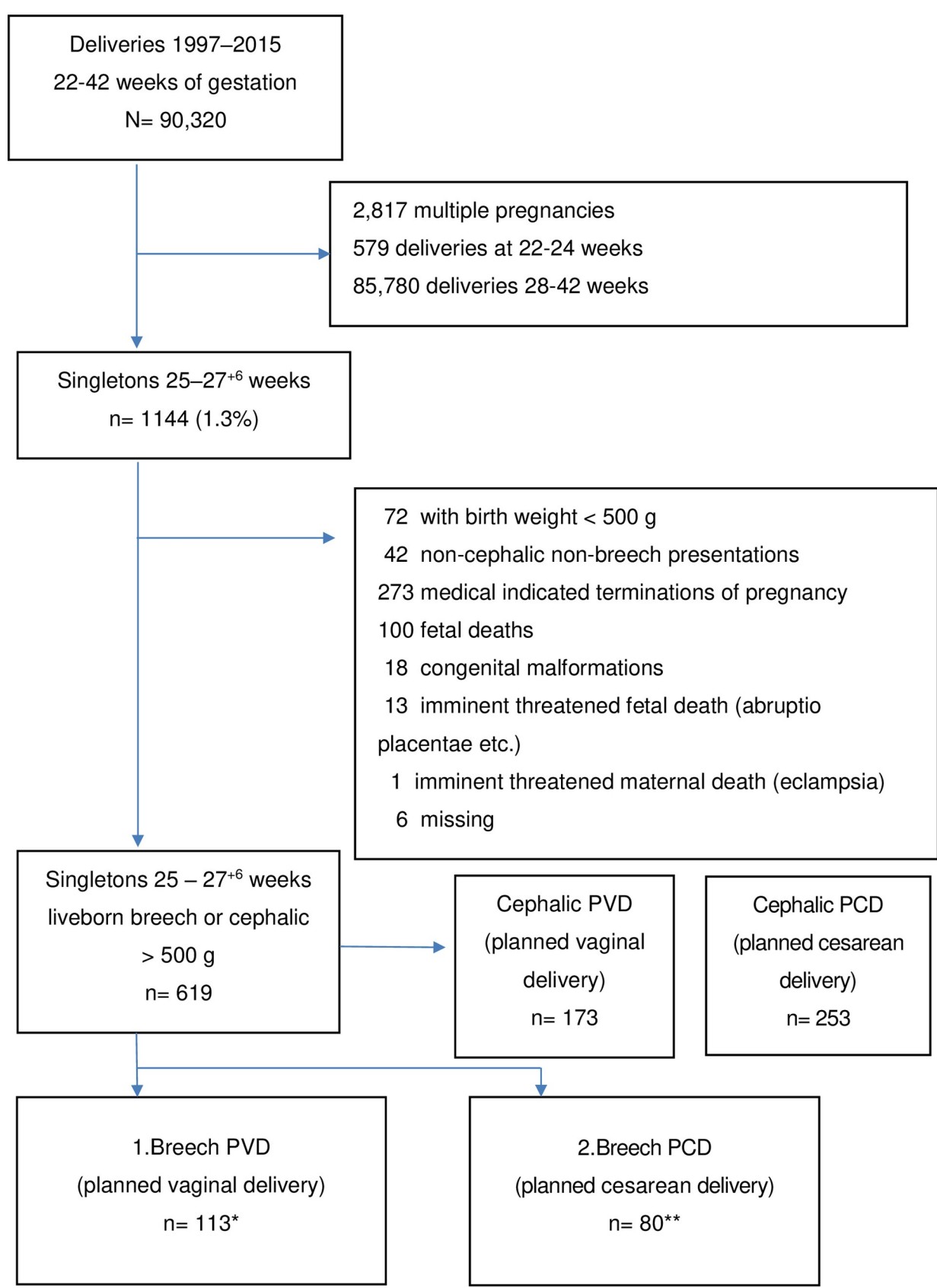

**Fig 1. Flow chart.** * including 4 cesareans during labor; ** none born by vaginal delivery.

**Table 1. Mothers' characteristics and pregnancy outcomes according to planned type of delivery (PVD: Planned vaginal delivery, PCD: Planned cesarean delivery).**

|  | 1. PVD | 2. PCD | P |
|---|---|---|---|
|  | n = 113 | n = 80 |  |
| Maternal age (years) | 27.3 ± 6.0 | 29.9 ± 5.5 | **0.002** |
| Smoker | 30/108 (27.8) | 13/78 (16.7) | 0.076 |
| BMI[1] (kg/m²) | 23.5 ± 5.3 | 25.1 ± 7.8 | 0.16 |
| Nulliparous | 62/113 (54.9) | 42/79 (53.2) | 0.82 |
| Previous cesarean section | 6/113 (5.3) | 18/79 (22.8) | <**0.001** |
| Spontaneous preterm delivery | 113/113 (100) | 26/80 (32.5) | <**0.001** |
| Preterm premature rupture of the membranes | 59/113 (52.2) | 22/80 (27.5) | <**0.001** |
| Antenatal corticosteroid therapy | 102/113 (90.3) | 77/80 (96.3) | 0.11 |
| Anesthesia/analgesia |  |  | < **0.001** |
| None | 61/111 (55.0) | 0/79 (0) |  |
| Locoregional | 50/111 (45.0) | 41/79 (51.9) |  |
| General | 0/111 (0) | 38/79 (48.1) |  |
| Duration of labor (min)² | 47.5 [10–110] | - | - |
| <60 minutes | 57/110 (51.8) | - | - |
| Gestational age at delivery (weeks) | 26.3 [25.6–26.9] | 27.1 [26.6–27.4] | < **0.001** |
| 25–25⁺⁶ | 39/113 (34.5) | 7/80 (8.7) | < **0.001** |
| 26–26⁺⁶ | 46/113 (40.7) | 25/80 (31.3) |  |
| 27–27⁺⁶ | 28/113 (24.8) | 48/80 (60.0) |  |
| Birth weight (grams) | 870 ± 130 | 760 ± 180 | <**0.001** |
| 500–699 | 9/113 (8.0) | 35/80 (43.7) | < **0.001** |
| 700–899 | 53/113 (46.9) | 28/80 (35.0) |  |
| ≥900 | 51/113 (45.1) | 17/80 (21.3) |  |
| Birthweight < 3rd percentile | 6 (5.3) | 42 (52.5) | <**0.001** |
| Sex ratio (male) | 75/113 (66.4) | 33/80 (41.3) | <**0.001** |

Values are expressed as mean ± standard deviation, median [interquartile range] or no./total no. (percentage).

[1]: 49 missing values (13 in PCD group), [2]: 3 missing values in PVD group.

these 619 women, 173 women had a cephalic PVD and 253 a cephalic PCD. Finally, 113 had a fetus in breech PVD—including 4 who finally had cesarean deliveries during labor (3.5%)—and 80 women had a breech PCD, all delivered according to plan. Among the planned cesareans, 88.2% performed incisions into the uterine body.

The characteristics of the women in the breech PVD and PCD group were very different (Table 1). In the PCD group, women were somewhat older and they had more often previous history of c-section. In this group, most of the deliveries were induced, gestational age was higher by several days, and the mean birth weight was more than 100 grams lower.

The neonatal conditions according to planned mode of delivery are described in Table 2. In the PVD group, more than a quarter of the infants presented at birth with either entrapment of the aftercoming head or difficult instrumental deliveries, significantly higher than in the PCD group (27.4 vs 6.3%, p<0.001, OR 5.7, 95% CI 2.1–15.3)) Forceps were clearly used more frequently in the breech PVD group than in the PCD group (12.4 vs 0.0%, p = 0.001). There was no difference between PVD and PCD groups according low 5-min Apgar scores. Among the children whose pH could be measured (n = 164, 85.0%), there was no significant difference between the groups in the number with severe neonatal acidosis (umbilical artery pH < 7.0), or with metabolic acidosis (base deficit > -12 mM/L), or with significantly elevated lactates. However, unadjusted analysis showed that total neonatal mortality in the breech PVD group

**Table 2. Newborn conditions according to planned mode of delivery.**

| | 1. PVD | 2. PCD | P | Unadjusted | Adjusted |
|---|---|---|---|---|---|
| | n = 113 | n = 80 | | OR [95% CI] | ORa [95% CI] * |
| Entrapment of the aftercoming head/difficulties during delivery | 31/113 (27.4) | 5/80 (6.3) | <0.001 | 5.7 [2.1–15.3] | |
| Use of forceps | 13/105 (12.4) | 0/80 (0.0) | 0.001 | NA | |
| 5-min Apgar < 7 | 17/95 (17.9) | 15/79 (19.0) | 0.85 | 0.9 [0.4–2.0] | |
| Umbilical artery pH[1] | 7.30 ± 0.12 | 7.25 ± 0.10 | 0.012 | 0.04 [0.01–0.08] | |
| < 7.0 | 2/96 (2.1) | 1/68 (1.5) | 1.00 | NA | |
| Base deficit > -12 mM/L | 5/89 (5.6) | 2/65 (3.1) | 0.46 | 1.9 [0.4–10.0] | |
| Lactates>10 mM/L on arrival at NICU | 7/88 (8.0) | 5/50 (10.0) | 0.68 | 0.8 [0.2–2.6] | |
| Umbilical artery < 7.0, base deficit > -12 mM/L or lactates>10 mM/L on arrival at NICU | 13 (11.5) | 8 (10.0) | 0.74 | 1.2 [0.4–3.5] | |
| Global neonatal mortality | 22/113 (19.5) | 6/77 (7.8) | 0.031 | 2.9 [1.1–7.4] | 2.6 [0.8–9.3] |
| • In the delivery room | 14/113 (12.4) | 0/77 (0.0) | 0.001 | NA | NA |
| • In the NICU | 8/113 (7.1) | 6/77 (7.8) | 0.85 | 0.9 [0.3–2.7] | 0.7 [0.2–2.9] |
| Maternal-fetal infection | 14/99 (14.1) | 10/77 (13.0) | 0.82 | 1.1 [0.5–2.6] | |
| Intraventricular hemorrhage grade 3–4 | 8/99 (8.1) | 9/76 (11.8) | 0.41 | 0.7 [0.2–1.8] | |
| Leukomalacia ≥ 3 or hyperechogenicity > 10 days | 17/99 (17.2) | 11/72 (15.3) | 0.74 | 1.2 [0.5–2.6] | |
| Intracranial hemorrhage | 8/98 (8.2) | 2/31 (6.5) | 0.76 | 1.3 [0.3–6.4] | |

Values are expressed as mean ± standard deviation or no./total no. (percentage). ORa: Adjusted for weight, gestational age and gender at delivery.

*: Difference mean were calculated as effect size for continuous variables. NA: not applicable.

[1]: 29 missing values (12 in PCD group).

was more than twice that in the breech PCD group (19.5 vs 7.8%, p = 0.031, OR 2.9, 95% CI [1.1–7.4], NNT = 8). This higher rate of neonatal mortality in the PVD group was associated exclusively with a higher risk of death in the delivery room (12.4 vs 0.0% *p* = 0.001, OR not calculable, NNT = 8). Finally, the newborns who left the delivery room alive for the NICU did not differ between the groups for neonatal mortality (7.1 vs 7.8%, p = 0.85, OR 0.9, 95% CI [0.3–2.7]) and neonatal severe morbidity: frequency of maternal-fetal infections, grade 3 or 4 intraventricular hemorrhages, periventricular leukomalacia of grade ≥ 3, hyperechogenicity > 10 days, or intracranial hemorrhages were identical in both groups.

The risk of death in the delivery room for breech PVDs remained stable around 12% over our 19-year study period: 18, 11, 20 and 10% during 5 years consecutive periods (chi2 for trend, Cochran-Armitage test, p = 0.62). Table 3 presents the factors associated with death in

**Table 3. Factors associated with death in the delivery room of infants born in breech planned vaginal delivery (breech PVD).**

| | Died in the delivery room | Alive at discharge from delivery room | P | OR [95% CI] |
|---|---|---|---|---|
| | n = 14 | n = 99 | | |
| Duration of labor < 60 min | 11/14 (78.6) | 46/99 (46.5) | 0.035 | 4.2 [1.1–16.1] |
| Cord prolapse | 3/14 (21.4) | 1/99 (1.0) | 0.006 | 26.7 [2.6–279.5] |
| Entrapment of the aftercoming head | 9/14 (64.3) | 22/99 (22.2) | 0.002 | 6.3 [1.9–20.7] |
| including cervicotomy | 6/14 (42.9) | 6/99 (6.1) | <0.001 | 11.6 [3.0–44.5] |
| Nulliparous women | 8/14 (57.1) | 54/99 (54.5) | 0.86 | 1.1 [0.4–3.4] |
| Preterm premature rupture of membranes | 10/14 (71.4) | 49/99 (49.5) | 0.13 | 2.6 [0.7–8.7] |
| Chorioamnionitis | 8/11 (72.7) | 51/68 (75.0) | 0.87 | 0.9 [0.2–3.7] |
| Complete breech | 8/14 (57.1) | 51/99 (51.5) | 0.69 | 1.3 [0.4–3.9] |
| Gestational age 25 weeks (vs 26 or 27) | 8/14 (57.1) | 31/99 (31.3) | 0.065 | 2.9 [0.9–9.1] |

Values are expressed as no./total no. (percentage). NA: not applicable

**Table 4. Risk of head entrapment and neonatal deaths in delivery room.**

|  | Head entrapment | Death in delivery room | Head entrapment with death in delivery room |
|---|---|---|---|
|  | **n = 31/113 (27.4)** | **n = 14/113 (12.4)** | **n = 9/31 (29.0)** |
| 25 SA (n = 39) | 12/39 (30.8) | 8/39 (20.5) | 6/12 (50.0) |
| 26 SA (n = 46) | 9/46 (19.6) | 5/46 (10.9) | 2/9 (22.2) |
| 27 SA (n = 28) | 10/28 (35.7) | 1/28 (3.6) | 1/10 (10.0) |
| p | 0.78 | **0.035** | **0.037** |

Values are expressed as no./total no. (percentage).

the delivery room among the 113 newborns in the breech PVD group. These deaths were significantly associated with a short duration of labor (<60mn) (78.6 vs 46.5%, p = 0.035, OR 4.2, 95% CI 1.1–16.1), cord prolapse (21.4 vs 1.0%, p = 0.006, OR 26.7, 95% CI 2.6–279.5) and entrapment of the aftercoming head (64.3 vs 22.2%, p = 0.002, OR 6.3, 95% CI 1.9–20.7). Cord prolapse, however, affected only 4 fetuses (3 of whom died in the delivery room), while more than a quarter of breech PVD fetuses had intracervical head entrapment (n = 31, 27.4%), and a third of them died in the delivery room (n = 9, 29.0%). Twelve of these cases of entrapment of the aftercoming head required the performance of a cervicotomy (10.6%), half of which were themselves accompanied by death in the delivery room (42.9 vs 6.1%, p<0.001, OR 11.6, 95% CI 3.0–44.5). On the other hand, deaths in delivery room were not significantly associated with nulliparity, premature rupture of the membranes, chorioamnionitis, type of breech, or a lower gestational age. Finally, among all deaths in delivery room for breech fetuses, head entrapments concern 9/14 and cord prolapse 3/14, risks known to be associated with breech presentation in vaginal delivery.

Table 4 reports the risk of head entrapment and death in delivery room according to gestational age in the cases of PVD. The risk of head entrapment barely changed with gestational age (mean: 27%). Death in delivery room fell as gestational age increased, and nearly all the neonatal deaths occurred in children born at 25 or 26 weeks (92.9%). Finally, the risk of death associated with head entrapment was significantly highest at the lowest gestational ages (50.0% at 25 weeks, 22.2% at 26 weeks, and 10.0% at 27 weeks).

## Discussion

Our study showed that among planned vaginal deliveries between 25 and 27 weeks, breech presentation was associated with neonatal mortality more than twice for that for breech presentations with PCD. It showed that the increased neonatal mortality for breech PVD infants we observed was associated with the excess of mortality in the delivery room.

This reduction in neonatal mortality with planned c-section before 28 weeks is accordingly consistent with the published data up to now. The reduction of crude neonatal mortality from 19.5% to 7.5% that we observed has been found in several series of extremely preterm fetuses, with gestational age limits a little different from those we chose. These reductions range from 25 to 13% in a large multicenter study between 24 and 28 weeks in the United States [7], 32 to 10% in a Swedish registry study of births between 25 and 28 weeks [10], from 44 to 24% in a retrospective single-center Canadian study of fetuses aged 23 to 25 weeks [12] from 55 to 20% in a study of infants born in Australia and New Zealand between 23 and 27 weeks [14], from 65 to 43% in infants born between 24 and 26 weeks in a single-center Australian study [11], and by half among infants born between 22 and 27 weeks in the Swedish EXPRESS cohort [15]. Two large retrospective multicenter studies in France also found reductions in neonatal mortality for fetuses with a somewhat higher gestational age, although these did not reach

statistical significance—from 11 to 7% and from 16 to 12% (26–29 weeks) [16, 17]. In a meta-analysis published in 2018, Grabovac et al. assembled data from 15 studies combining 12,335 infants in breech presentation with PVD between 23 and 27 weeks and observed a significant diminution of the risk of neonatal mortality for cesarean deliveries [13].

One of the criticisms that can be made of studies that compared breech presentations with PVD with breech by PCD [11, 13–15] is that PVD are the culmination of spontaneous labor, while those by PCD include a high proportion of induced deliveries, which is a very different obstetric context. Furthermore, the reason why 26 of the 139 fetuses (18.7%) whose labor was spontaneous had a PCD and not a PVD in our study was not always known. Our results, however, are consistent with other studies. Two studies argue in favor of a specific risk of PVD in case of breech presentation, that is not found in case of cephalic presentation. In a multivariate analysis of more than 1000 fetuses born in the USA between 24 and 27 weeks, Reddy et al. observed that the risk of mortality tripled among the breech PVDs compared with the breech PCDs, although they observed no significant increase in neonatal mortality between cephalic presentations with PVD and with PCD [7]. Similarly in the Australian series by Thomas et al., the doubled rate of neonatal mortality observed in preterm breech presentations between 24 and 26 weeks was not found among the infants with cephalic presentations [11]. These data are therefore consistent and tend to indicate that the excess mortality of extremely preterm infants in breech presentations is indeed associated with the presentation.

Until now, none of the studies examining neonatal mortality in extremely preterm infants in breech presentation have differentiated deaths in the delivery room from deaths in the NICU [9, 11–17]. In our study, the mortality of the newborns transferred to the NICU after birth was identical in both groups, thus corresponding to the mortality associated with this extreme preterm delivery. This shows that the increased neonatal mortality for breech PVD infants we observed was associated exclusively with the excess of mortality in the delivery room. In these vaginal births with death in delivery room, head entrapments concern 9/14 and cord prolapse 3/14: that is, 12 out of 14 were associated with risks known to be related to breech presentation in vaginal delivery. Thus, mortality in the delivery room appears to be associated mainly with an excess number of complications that are strongly related to breech presentations [21]. Although these risks have been mentioned by others [14, 16, 17], our study is original in highlighting the cause of this excess mortality in the delivery room.

Our study has several limitations. First, the groups compared retrospectively are far from identical in terms of gestational age and birth weight (aged a few days extra, but very often severely stunted in the PCD group). This is common to all retrospective studies of this type [14, 15, 19]. Secondly, our retrospective study covers a 19-year period in a single center, with changes in practices and neonatal results during this period. Nonetheless, the absence of variation in the risk of death in the delivery room of fetuses in breech presentation during the study period argues in favor of a risk specific to breech presentation. Moreover, to limit as much as possible the bias associated with changes in the practices for active management, we limited our analysis to fetuses born at and after 25 weeks of gestation and weighing more than 500 grams. In our center, these children have systematically received active management, as evidenced by the rate of antenatal corticosteroid therapy of at least 90% in each group.

Although no guidelines currently recommend a specific mode of delivery for extremely preterm infants in breech presentation, our study finally militates in favor of their systematic cesarean delivery. When labor occurs spontaneously, the decision about a cesarean must be made rapidly since the duration of labor is very short. A cesarean at this very early gestational age exposes the woman to a high risk of well-known complications [22, 23] especially to a scar in the uterine body in most cases (88% of the cesareans in our study) and to an excess risk of maternal mortality [24–26]. Maternal morbidity is also substantial [6, 26–28] with

complications in the short term (postpartum hemorrhages and severe anemia, thromboembolic and infectious events) and the long term—for subsequent pregnancies (uterine rupture, preterm delivery, abnormalities of placentation). The balance between benefits and risks—high for both the fetus and the mother—must be evaluated and considered. For the fetal risks, our study shows that a cesarean may avoid a neonatal death every 8 interventions. This evidence is important for professionals but also for parents, who must always be informed as clearly as possible about each conceivable option. It is important to recall that in extremely preterm births, a systematic cesarean is part of active management of fetuses at the limit of viability, to increase their survival and decrease the risk of long-term sequelae [5, 15, 19, 29, 30]. Finally, our study shows a cesarean could be systematically proposed for breech presentations before 28 weeks of gestation.

## Author Contributions

**Conceptualization:** Clémentine Pierre, Audrey Leroy, Laurent Storme, Sandrine Depret, Thameur Rakza, Charles Garabedian.

**Data curation:** Clémentine Pierre.

**Formal analysis:** Clémentine Pierre, Adeline Pierache.

**Investigation:** Clémentine Pierre, Véronique Debarge.

**Methodology:** Clémentine Pierre, Charles Garabedian.

**Software:** Clémentine Pierre.

**Supervision:** Thameur Rakza.

**Validation:** Clémentine Pierre.

**Visualization:** Clémentine Pierre.

**Writing – original draft:** Clémentine Pierre, Damien Subtil.

**Writing – review & editing:** Clémentine Pierre, Damien Subtil.

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
