## [Decision Letter · Decision Letter 0]

29 Dec 2020

PONE-D-20-34255

Is vaginal delivery of a fetus in breech presentation at an extremely preterm gestational age associated with an increased risk of neonatal death? A comparative study

PLOS ONE

Dear Dr. pierre,

Thank you for submitting your manuscript to PLOS ONE. After careful consideration, we feel that it has merit but does not fully meet PLOS ONE’s publication criteria as it currently stands. Therefore, we invite you to submit a revised version of the manuscript that addresses the points raised during the review process.

Two experts in the field handled your manuscript, and we are grateful for their time and contributions. Although interest was found in your study, numerous concerns arose that require your attention. Please address ALL of the reviewers' comments in your revised manuscript.

We look forward to receiving your revised manuscript.

Kind regards,

Frank T. Spradley

Academic Editor

PLOS ONE

2. In the ethics statement in the manuscript and in the online submission form, please provide additional information about the patient records/samples used in your retrospective study, including:

a) whether all data were fully anonymized before you accessed them;

b) the date range (month and year) during which patients' medical records/samples were accessed;

c) details on the mechanism of the opt-out consent method; and

d) the source of the medical records/samples analyzed in this work (e.g. hospital, institution or medical center name).

3. Please include details regarding CNIL authorization in the manuscript Methods.

4. Please amend the manuscript submission data (via Edit Submission) to include authors Audrey Leroy, Adeline Pierrache, Laurent Storme, Véronique Debarge, Sandrine Depret, Thameur Rakza, Charles Garabedian and Damien Subtil.

Reviewers' comments:

Reviewer's Responses to Questions

**Comments to the Author**

1. Is the manuscript technically sound, and do the data support the conclusions?

Reviewer #1: Partly

Reviewer #2: Partly

2. Has the statistical analysis been performed appropriately and rigorously? 

Reviewer #1: No

Reviewer #2: Yes

3. Have the authors made all data underlying the findings in their manuscript fully available?

Reviewer #1: No

Reviewer #2: Yes

4. Is the manuscript presented in an intelligible fashion and written in standard English?

Reviewer #1: Yes

Reviewer #2: Yes

5. Review Comments to the Author

Reviewer #1: Thanks you for giving me the opportunity to review this mansucript

How should preterm breech fetus be planned to deliver ? This is an intersting topic. Given that it is not a frequent situation, all experiences are welcome to counsel women.

In fact, this mansucript is incomplete and flawed by many methodologic bias that need to be adressed before publication.

Moreover, the text is not well structured because there are an amount of data and the results should be separated.

Therefore, the manuscript IS VERY SPRAWLING (breech versus cephalic, factors associated with mortality in vaginal delivery, explanation of risk of entrapment ....)

Results of this manuscript could be interesting if the authors performed several changes so that readers could undertsand and compared the results to their own data.

See below the main comments

1. The aim of the study is to compare Planned vaginal delivery and planned caesarean delivery for preterm breech delivery. Therefore, the aim IS NOT to study if preterm breech delivery has an increased risk of perinatal death.

One may suppose that breech fetus are at an increased risk of perinatality. The increased risk observed in these fetuses coule be linked to antenatal or underlying disorders that may be associated with the breech presentation and so not solely due to the mode of delivery. In that way, it is intersting to compare cephalic versus breech presentation.

However it is not your topic.

When you councel a women, you are not going to councel her by saying : as compared to cephalic presention, breech presentation is at increased risk, so let's perform a CS.

You are going to explain , by comparing the MODE of delivery for preterm breech delivery, the safer is ... or is not..

Moreover, by excluding congenital malformation, birthweight < 500g, medical termination of pregnancy and fetal detah, if the ratio cephalic/breech is not equally distributed as compared to the selected population, selection bias is highly probable. As I could read : 69,1% of preterm cephalic had CS versus 31,25 % of preterm breech . This important difference can not be a random ratio. It reflects selection bias that may favor cephalic presentation

Therefore, in my opinion group 2 is not necessary. Only data on the overall neonatal mortality rate of preterm cephalic fetus is necessary to give to have an overview of the practice in your center

2. Period study : authors should explained why they ended the inclusion to the years 2015

3. Main problem of external validity that need to be adressed

- the group 1 :CS rate during labor 3,5%

This is a french study. I could compare this results with 2 recent french studies : Kayem et al (26-30 WP) (20%) and lorthe (24-34 WP)et al (18,4%)

How can the authors explain this important difference ?

External validity is questionable .. ?

4. As written in the text, the authors argued for a high experience in breech delivery. Therefore in the method, authors should give the selection criteria for a trial of labor

5. Period study. Between 1997-2015, several pediatrics improvments have been performed . Authors performed a chi-square test by comparing neonatal mortality betwwen 1997-2005 and 2005-2015. I am not convinced by this method that is very biased and unprecised.Therefore to be sure that there is no time varying effect, authors should at least give :

- A run line series plot for the 2 groups

- And if you can Check for trends and then you can perform a correlation test (not useful for some statisticien), or computing the variance of the 2 groups if they are independent (not reasonable assumption, because it is the same hospital ?) or Vector Auto-regression model if the 2 groupes are dependant

6. The authors found an increased risk of neonatal morbidity in planned vaginal delivery. What we want to know is : why ? Is it solely the mode of delivery or the obstetrical underliyng condition or selection bias ?

Missing important data : reason for CS DELIVERY ? Reason for preterm delivery ?

According to recent french data , (delorm et al. 2016), cause of preterm delivery could be a prognotic factors (such as low birthweight...) .

As I can read in the table 1, in the group 3 (PCD), 32,5% had spontaneous preterm delivery. What was the reason to perform CS ?

Therfore the main problem is the allocation to PVD or PCD ? This is not well described in the manuscript

Due to its retrospective nature the authors should be very precise to describe the reason for the mode of delivery

7- As written in the results, group 1 and 3 are very different. Therefore, all these confounding variables need to be addressed in a regression analysis

8- Discrepencies in the data : Latter deliveries but lower Birthweight in PCD ... This may reflect either selection bias or codding bias or different policies for planned dleivery that need to be explained.

9- Duration of labor : less than 60min in 50%. Authors should explained how they defined the beginning of the labor.

10- missing data, that need to be adressed in group 1 and 3 : type of breech (franck, non franck, footling), clinical chorioamnionitis, use of tocolysis, the use of magnesium sulphate, economic status, admission after labor onset (in utero transfer), sex ratio, use of antibiotics, detailed of breech vaginal delivery management

11- Regression analysis should be detailed. In the table 2 (with some form typo), I could read adjusted OR. However I could not see either in the method nor in the result section the explanation of this adjustment.

Factors of death have been studied only in the PVD group?

Usually to perform adjused analysis, you should select factor according :

*confounding factors

*no colliders

* avoiding unnecessary covariate

* avoiding multicolinearity

And then you run a regression

12. Table 3-4 : This table adresses the factors associated with mortality in PVD. This is very intersting and could be a rational for another manuscript.

In conclusion, due to the amount of data, I suggested to the authors :

- To study mode of delivery on breech fetus : PVD vs PCD. To perform a classical analysis of the causal framework between mode of delivery and mortality

- Or to to study only group 1 and to perform an analysis (by regression, or a detailed descriptive analysis) to understand the neonatal mortality after PVD

Reviewer #2: Pierre and colleagues compare neonatal outcomes along three groups of singleton preterm deliveries: planned breech vaginal delivery, planned cephalic vaginal delivery, planned breech cesarean delivery.

The authors did a nice job in the discussion explaining how their research adds to the literature (consideration of spontaneous vs induced preterm delivery; distinction between death in delivery room vs NICU). The introduction would be strengthened by similarly setting up a distinction between what is known from prior literature and what this study adds.

My biggest concern is the difference in findings between the bivariate analysis and adjusted analysis, how these findings are explained, and the conclusions drawn from them. The bivariate analysis demonstrates that neonatal mortality in the breech PVD group was twice that of the cephalic PVD group (OR = 2.5) and almost triple that of the breech PCD group (OR = 2.9). However, as reported, there are significant differences in gestational age, birthweight, and rates of induction between the breech vaginal and breech c-section groups. After adjusting for GA and birthweight, there is no longer a significant difference in morbidity between the breech vaginal and breech c-section groups. Gestational age is the known key factor in outcomes for premature deliveries, so this is not unexpected. In the abstract, the highlighted results should be the adjusted ORs. The conclusions drawn should be that differences in neonatal mortality between breech vaginal and breech c-section groups were no longer significant after controlling for gestational age and birthweight (although differences between cephalic PVD and breech PVD groups persist). It is also unclear if the adjusted OR are adjusted for variables other than GA and birthweight? This should be described clearly. Other clinically relevant variables like induction, parity, receipt of steroids, presence of chorio, should be evaluated in bivariate analysis and considered for inclusion in the adjusted logistic regression.

The discussion should comment on why the authors think the GA is higher in the PCD groups but the birthweight is lower.

If data on sex of the neonates is available, that could be added to the adjusted model as a control. Male fetuses are known to have worse outcomes, and sex is often included in outcome prediction (https://www.nichd.nih.gov/research/supported/EPBO/use).

Regarding decisions to deliver vaginally vs by c-section, it would be helpful to explain in the methods how PVD vs PCD is determined from record review. Was there any hospital protocol to make this decision, or did the attending obstetrician use their personal clinical judgement? A context about the number of different providers making these decisions would also be helpful. Since this is an observational retrospective study, this study is limited by non-measured factors (potential confounds) that made healthcare providers decide to deliver some of these breech fetuses vaginally and others by c-section. This should be added to the discussion.

The term “global” neonatal mortality is confusing. I recommend changing it to “total” and defining it clearly as death during delivery + death in NICU. Was NICU followup stopped at 28 days (definition of neonatal mortality) or continued though discharge?

Table 3: What is “Gestational age 25 weeks” compared to? Is this 25/0-25/6 compared to >26 weeks? The title is confusing.

Table 4: The title is confusing. Categories with <5 items should be compared using Fisher’s Exact, not Chi Squared. In “25 SA”, “26 SA”, etc, what does “SA” mean?

6. PLOS authors have the option to publish the peer review history of their article (what does this mean?). If published, this will include your full peer review and any attached files.

Reviewer #1: No

Reviewer #2: No

---

## [Author Response · Author response to Decision Letter 0]

26 May 2021

We thank the reviewers for their comments and questions which we answered in the attached "response to reviewer" file attached at the manuscript

---

## [Decision Letter · Decision Letter 1]

15 Jun 2021

PONE-D-20-34255R1

Is vaginal delivery of a fetus in breech presentation at an extremely preterm gestational age associated with an increased risk of neonatal death? A comparative study

PLOS ONE

Dear Dr. pierre,

Thank you for submitting your manuscript to PLOS ONE. After careful consideration, we feel that it has merit but does not fully meet PLOS ONE’s publication criteria as it currently stands. Therefore, we invite you to submit a revised version of the manuscript that addresses the points raised during the review process.

We look forward to receiving your revised manuscript.

Kind regards,

Frank T. Spradley

Academic Editor

PLOS ONE

Journal Requirements:

Reviewers' comments:

Reviewer's Responses to Questions

**Comments to the Author**

1. If the authors have adequately addressed your comments raised in a previous round of review and you feel that this manuscript is now acceptable for publication, you may indicate that here to bypass the “Comments to the Author” section, enter your conflict of interest statement in the “Confidential to Editor” section, and submit your "Accept" recommendation.

Reviewer #1: All comments have been addressed

Reviewer #2: (No Response)

2. Is the manuscript technically sound, and do the data support the conclusions?

Reviewer #1: Partly

Reviewer #2: Yes

3. Has the statistical analysis been performed appropriately and rigorously? 

Reviewer #1: Yes

Reviewer #2: Yes

4. Have the authors made all data underlying the findings in their manuscript fully available?

Reviewer #1: No

Reviewer #2: Yes

5. Is the manuscript presented in an intelligible fashion and written in standard English?

Reviewer #1: Yes

Reviewer #2: Yes

6. Review Comments to the Author

Reviewer #1: Some comments

Thanks you for giving me to review the revised manuscript. I am convinced that the manuscript could deserve publication. However to improve the clarity, the understanding and in fact external validity some comments need to be addressed

Some data needs to be published especially the excess of neonatal death due to head entrapment for example. In my opinion the authors should just simplified some analysis, the table 2 and the process of selection of women.

1. As the primary outcomes is the combined neonatal death in delivery room and in NICU, there is a little discrepencies between the sentences in the introduction and in the material and method

In order to better understand the possible excess mortality in case of trial of labor, we wanted to differentiate the mortality occurring in the delivery room from that occurring in the neonatal intensive care unit.

The principal endpoint was total neonatal defined as i) death during delivery plus ii) death in the neonatal intensive care unit (NICU).

2. I read the manuscript for a second time and I try to be a "new reader". I may not understand fully the objectives of the authors.

Firstly , The authors want to provide data to counsel women when : (i) preterm delivery is occuring (ii) fetus is in breech.

Therefore, the group of planned CS should not be a mixture of induce/spontaneous labor and real planned CS. Women with induction of labor should be in the planned vaginal birth

In fact, women with spontaneous preterm labor in breech or with premature preterm rupture of membrane or with medically indication birth should be offered 2 options: PVD or PCD = PVD (spontaneous or induction) vs PCD

So the authors should clearly explained the 26 women with spontaneous labor in the PCD group. I may imagine that it is women with spontaneous preterm labor and the obstetrician may have chosen to perform CS. However, should these women include in PCD group? May be yes in latent phase of labor, no in active phase of labor

Secondly the difficulty is that the group of PVD might be a group with worse prognosis. Therefore, the reader need to understand why the obstetrician chose PVD or PCD

3. Reason for neonatal morbidity should be given as no difference was observed between APGAR, chorioamniotis, IVH, hemorrhage, leukomalacia ?? What is the effect of antenatal corticosteroids?

4. Statistical analysis:

I am not convinced by the adjusted OR because some factors might be effect modifier

One simple possibility to understand the factors associated with neonatal mortality is to perform stratified analysis and mantel haenzel test to distinguish confounding factors and effect modifier: especially for gestational age, use of antenatal corticosteroids, gender, birthweight (500 – 600 – 700 – 800 or < 3°percentile vs no), by providing crude and adjusted M-H OR. In fact multiple logistic regression is somewhat not useful with small sample size and collinearity (birthweight and gestational age ++).

5. Can the authors explained the increased death in delivery room with cord prolapse ? Cord prolapse during expulsive stage or during active phase labor. As only 3.4% of women in PCD had CS , can the authors explaine the management according to cord prolapse?

6. The significance of this excess of mortality disappeared after adjusting for gestational age, birthweight and gender at delivery

This sentence should be deleted as statistical significance is trivial when analyzing neonatal death. In 2016, the American Statistical Association released a statement in The American Statistician warning against the misuse of statistical significance and P values

Not all values inside are equally compatible with the data, given the assumptions. The point estimate is the most compatible, and values near it are more compatible than those near the limits

In your manuscript, whatever the OR, there is a clear association between mode of delivery and neonatal death. The only question is why ?

In fact the authors should clearly explained the management during labor, and the other comment previously written (reason for PCD, effect modifier or confounding, discrepancies between no diiference for pH and other morbidity while neonatal death in delivery room only occurred after PVD)

7. Our study was carried out in a university maternity which performs around 5,000 deliveries per year. The selection criteria for a trial of labor has been the subject of a protocol and regular publications. Adjaoud S, J Gynecol Obstet Biol Reprod 2017 ; 46 :445-8.

I disagree with the authors. In this article, the protocol is for term breech trial and not for preterm delivery…..

I cited : « Women were included if they had given birth during the study period to a singleton child at term, in either breech or vertex presentation. Multiple pregnancies, medically indicated terminations of pregnancy, and in utero deaths were excluded from the study »

8; Table 2 :

typo 10 and 12% instead of 0,10 et 0,12.

I am not convinced with the p value for mean pH

Item asphyxia should be given i.e : ph < 7 or base excess or lactate

Reviewer #2: Overall, the authors did a very nice job with revisions and the paper is stronger and more statistically accurate.

Continued minor issues:

In the abstract, the conclusion states that “For deliveries between 25+0 and 27+6 weeks' gestation, vaginal delivery in breech presentation is associated with a significantly higher risk of death in the delivery room.” Again, I would be careful with using this language of “significance” when differences in mortality were significant in the adjusted analysis.

I understand the rationale of not adjusting for many variables given the small sample size, but I do think that parity (history of vaginal delivery) is a clinically very important variable when it comes to head entrapment risk and successful vaginal delivery. If you choose not to adjust for parity, I recommend specifically mentioning that as a limitation in the discussion.

In the third paragraph of the results, I recommend specifying that the initial results you describe are “bivariate analysis” or “unadjusted analysis.”

Adding a column to table 2 with adjusted OR (with a footnote explaining what is was adjusted for) may help readers better understand the difference in OR/significance for mortality in bivariate analysis vs adjusted analysis.

7. PLOS authors have the option to publish the peer review history of their article (what does this mean?). If published, this will include your full peer review and any attached files.

Reviewer #1: No

Reviewer #2: No

---

## [Author Response · Author response to Decision Letter 1]

24 Aug 2021

In the revisions requested by the reviewers, we have responded in the file "responses to reviewers" join to the revised version of the manuscript

---

## [Decision Letter · Decision Letter 2]

24 Sep 2021

Is vaginal delivery of a fetus in breech presentation at an extremely preterm gestational age associated with an increased risk of neonatal death? A comparative study

PONE-D-20-34255R2

Dear Dr. pierre,

We’re pleased to inform you that your manuscript has been judged scientifically suitable for publication and will be formally accepted for publication once it meets all outstanding technical requirements.

Kind regards,

Frank T. Spradley

Academic Editor

PLOS ONE

Reviewers' comments:

Reviewer's Responses to Questions

**Comments to the Author**

1. If the authors have adequately addressed your comments raised in a previous round of review and you feel that this manuscript is now acceptable for publication, you may indicate that here to bypass the “Comments to the Author” section, enter your conflict of interest statement in the “Confidential to Editor” section, and submit your "Accept" recommendation.

Reviewer #1: All comments have been addressed

Reviewer #2: All comments have been addressed

2. Is the manuscript technically sound, and do the data support the conclusions?

Reviewer #1: Yes

Reviewer #2: Yes

3. Has the statistical analysis been performed appropriately and rigorously? 

Reviewer #1: Yes

Reviewer #2: Yes

4. Have the authors made all data underlying the findings in their manuscript fully available?

Reviewer #1: Yes

Reviewer #2: Yes

5. Is the manuscript presented in an intelligible fashion and written in standard English?

Reviewer #1: Yes

Reviewer #2: Yes

6. Review Comments to the Author

Reviewer #1: The authors provided answers to the comments

The article can be published

In that form the article is short and more concise

Reviewer #2: Overall, the authors did a very nice job with revisions. I have no further concerns or suggestions for improvement. The paper has interesting and clinically relevant findings.

7. PLOS authors have the option to publish the peer review history of their article (what does this mean?). If published, this will include your full peer review and any attached files.

Reviewer #1: No

Reviewer #2: No

---

## [Editor Report · Acceptance letter]

11 Oct 2021

PONE-D-20-34255R2 

Is vaginal delivery of a fetus in breech presentation at an extremely preterm gestational age associated with an increased risk of neonatal death? A comparative study 

Dear Dr. pierre:

I'm pleased to inform you that your manuscript has been deemed suitable for publication in PLOS ONE. Congratulations! Your manuscript is now with our production department. 

Kind regards, 

on behalf of

Dr. Frank T. Spradley 

Academic Editor

PLOS ONE